# A Systematic Review and Meta-Analysis of the Effect of Statins on Glutathione Peroxidase, Superoxide Dismutase, and Catalase

**DOI:** 10.3390/antiox10111841

**Published:** 2021-11-19

**Authors:** Angelo Zinellu, Arduino A. Mangoni

**Affiliations:** 1Department of Biomedical Sciences, University of Sassari, 07100 Sassari, Italy; azinellu@uniss.it; 2Discipline of Clinical Pharmacology, College of Medicine and Public Health, Flinders University, Bedford Park, SA 5042, Australia; 3Department of Clinical Pharmacology, Flinders Medical Centre, Southern Adelaide Local Health Network, Bedford Park, SA 5042, Australia

**Keywords:** statins, glutathione peroxidase, superoxide dismutase, catalase, oxidative stress, pleiotropic effects

## Abstract

Statins may exert protective effects against oxidative stress by upregulating specific antioxidant mechanisms. We conducted a systematic review and meta-analysis of the effect of statins on three key antioxidant enzymes: glutathione peroxidase (GPx), superoxide dismutase (SOD), and catalase. The electronic databases PubMed, Web of Science, and Scopus were searched from inception to July 2021. The risk of bias was assessed with the Joanna Briggs Institute Critical Appraisal Checklist and certainty of evidence was assessed using the GRADE framework. In 15 studies, reporting 17 treatment arms in 773 patients (mean age 53 years, 54% males), statins significantly increased the concentrations of both GPx (standardized mean difference, SMD = 0.80, 95% confidence interval, CI 0.13 to 1.46, *p* = 0.018; high certainty of evidence) and SOD (SMD = 1.54, 95% CI 0.71 to 2.36, *p* < 0.001; high certainty of evidence), but not catalase (SMD = −0.16, 95% CI −0.51 to 0.20, *p* = 0.394; very low certainty of evidence). The pooled SMD values were not altered in sensitivity analysis. There was no publication bias. In conclusion, statin treatment significantly increases the circulating concentrations of GPx and SOD, suggesting an antioxidant effect of these agents (PROSPERO registration number: CRD42021271589).

## 1. Introduction

Elevations in circulating cholesterol concentrations significantly increase the risk of atherosclerosis and its clinical manifestations, particularly myocardial infarction, ischemic stroke, and peripheral arterial disease [1,2]. Statins, through the inhibition of the enzyme 3-hydroxy-3-methylglutaryl-CoA (HMG-CoA) reductase, the rate-limiting step in the mevalonate pathway through which cells synthesize cholesterol, are the most commonly prescribed drugs for the treatment of hypercholesterolaemia and the management of cardiovascular risk worldwide in view of their favourable efficacy and safety profile [3]. However, while the main action is mediated by lowering the concentrations of specific cholesterol fractions, particularly low-density lipoprotein (LDL) [4], the atheroprotective effects of statins involve other mechanisms, normally described as pleiotropic effects [5,6,7]. Such effects, generally apparent shortly after commencing statin treatment, have been shown to be mediated by specific antioxidant mechanisms [8,9,10,11].

Oxidative stress, through the generation of reactive oxygen species (ROS) and oxidized LDL, is considered to play a key pathophysiological role in the onset and the progression of atherosclerosis [12,13,14]. Specifically, oxidative stress exerts significant negative effects on cellular homeostasis by damaging lipids, thiols, DNA, and protein pools, stimulating the synthesis and release of pro-inflammatory and atherogenic cytokines, and favouring the adhesion of monocytes to the endothelium, a critical pathophysiological step in atherosclerosis and plaque formation [15,16]. The coexistence of oxidative stress and hypercholesterolemia imposes a particularly high burden on endothelial integrity, further increasing the risk of atherosclerosis and its clinical manifestations [17,18].

The effects of statin treatment, singly or in combination with other therapies, on oxidative stress have been extensively studied both in experimental models of atherosclerosis and in humans [19,20,21]. In particular, statins have been shown to inhibit key pro-oxidant enzymes such as nicotinamide adenine dinucleotide phosphate (NADPH) oxidase [22,23], reduce the synthesis of the highly reactive compound malondialdehyde from lipid peroxidation of polyunsaturated fatty acids [24], as well as increase the expression, activity, and coupling of endothelial nitric oxide synthase [25], and upregulate antioxidant enzymes such as catalase [26], glutathione peroxidase (GPx) [27], and superoxide dismutase (SOD) [28,29,30]. Notably, epidemiological studies have convincingly shown that higher circulating concentrations of the antioxidant enzymes GPx, SOD, and catalase are associated with a significant reduction in the risk of coronary heart disease [31]. This suggests that pharmacological strategies that upregulate these enzymes may exert a key protective role against atherosclerosis and cardiovascular disease.

In order to investigate the complex interplay between statins and antioxidant mechanisms, we conducted a systematic review and meta-analysis of studies reporting on the effects of statin treatment on the circulating concentrations of GPx, SOD, and catalase in patients with different cardiovascular risk profiles. We hypothesised that statin treatment would significantly increase GPx, SOD, and catalase concentrations regardless of specific agents used.

## 2. Materials and Methods

### 2.1. Search Strategy and Study Selection

We searched for articles published in PubMed, Web of Science, and Scopus, from inception to 31 July 2021, using the terms “Glutathione Peroxidase” or “GPx” or “GSH-PX” or “Superoxide Dismutase” or “SOD” or “Catalase” and “Statin”. The abstracts and articles were screened by two independent investigators. The article references were also searched for additional studies. Pre-defined inclusion criteria were: (a) reporting of GPx and/or SOD and/or catalase concentrations in blood, erythrocytes, plasma, or serum at baseline and after statin treatment; (b) ≥10 participants; (c) English language; and (d) full-text availability. Data extracted included the country, type of biological matrix, age, sex distribution, GPx, SOD, and catalase concentrations before and after treatment, disease condition studied, statin and dose used, and treatment duration.

The Joanna Briggs Institute (JBI) Critical Appraisal Checklist for analytical studies was used to assess the risk of bias. Scores ≥ 5, 4, and <4 indicated low, moderate, and high risk, respectively [32]. The Grading of Recommendations, Assessment, Development and Evaluation (GRADE) working group system was used to assess the certainty of evidence. GRADE considers the study design, the risk of bias, the presence of heterogeneity, the indirectness of evidence, the imprecision of results, the effect size (small, SMD < 0.5, medium, SMD 0.5–0.8, and large, SMD > 0.8) [33], and the publication bias [34,35,36]. The study was conducted in accordance with the Preferred Reporting Items for Systematic Reviews and Meta-Analyses (PRISMA) 2020 statement on the reporting of systematic reviews and meta-analyses (Appendix A) [37]. The International Prospective Register of Systematic Reviews (PROSPERO) registration number was CRD42021271589.

### 2.2. Statistical Analysis

Because of the different units of measurement (U/mL, U/gHb, nmol/mg, or µmol/L) used to express the concentrations of GPx, SOD, and catalase, standardized mean differences (SMDs) and 95% confidence intervals (CIs) were calculated to build forest plots of the differences in GPx, SOD, and catalase concentrations before and after statin treatment, with a *p*-value < 0.05 indicating statistical significance. When required, the means and standard deviations were derived from the corresponding medians and interquartile ranges [38], medians and ranges [39], or from graphs using the Graph Data Extractor software. Between-study heterogeneity was assessed using the Q-statistic (significance level set at *p* < 0.10) and the I^2^ statistic (I^2^ < 25%, no heterogeneity; I^2^ = 25–50%, moderate heterogeneity; I^2^ = 50–75%, large heterogeneity; I^2^ > 75%, extreme heterogeneity) [40,41]. In the presence of significant heterogeneity, defined as I^2^ values ≥ 50%, a random-effects model was used. Sensitivity analysis was performed to assess the influence of each study on the overall risk estimate by sequentially removing individual studies [42]. Publication bias was assessed with the Begg’s test, the Egger’s test (significance level set at *p* < 0.05 for both), and the “trim-and-fill” procedure [43,44,45]. When possible, the effects of individual statins (e.g., lipophilic: atorvastatin, simvastatin, lovastatin, fluvastatin, cerivastatin, and pitavastatin; hydrophilic: rosuvastatin, pravastatin) were assessed and compared. Statistical analyses were performed using Stata 14 software (STATA Corp., College Station, TX, USA).

## 3. Results

### 3.1. Study Selection

We initially identified 1988 articles. A total of 1970 were excluded (duplicates or irrelevant). After reviewing the remaining 18 articles, 3 were further excluded, leaving 15, reporting 17 treatment arms in 773 patients (mean age of 53 years, 54% males), for final analysis (Figure 1 and Table 1) [27,46,47,48,49,50,51,52,53,54,55,56,57,58,59].

### 3.2. Glutathione Peroxidase

#### 3.2.1. Study Characteristics

A total of 13 studies, reporting 14 treatment arms in 558 patients (mean age 54 years, 52% males), presented data on GPx concentrations [27,46,47,48,49,50,51,52,53,54,55,57,58]. Erythrocytes were assessed in five studies (six arms) [27,48,50,51,54], whole blood in two [52,53], serum in four [47,49,57,58], and plasma in the remaining two [46,55]. The statin used was atorvastatin in four studies [49,50,51,58], simvastatin in four [47,48,51,57], fluvastatin in three [27,52,54], and pravastatin [46] and rosuvastatin [53] in one, respectively. Treatment duration ranged between four weeks and three years (Table 1).

#### 3.2.2. Risk of Bias

The risk of bias was low in 11 studies [27,46,48,49,50,51,53,54,55,57,58] and high in the remaining 2 [47,52] (Table 2).

#### 3.2.3. Results of Individual Studies and Syntheses

The forest plot of the GPx concentrations before and after statin treatment is shown in Figure 2. In 10 treatment arms [27,46,48,49,51,52,53,54,57], the circulating GPx concentrations were higher after statin treatment (mean difference range, 0.18 to 4.50), with a significant difference reported in six [27,48,49,51,54]. By contrast, in four arms [47,50,55,58], the GPx concentrations were lower after treatment (mean difference range, −0.13 to −0.60), with a significant difference reported in one [50]. Random-effects models were used in view of the extreme heterogeneity observed (I^2^ = 96.0%, *p* < 0.001). Pooled results showed that circulating GPx concentrations were significantly higher after statin treatment (SMD = 0.80, 95% CI 0.13 to 1.46, *p* = 0.018). In sensitivity analysis, the corresponding pooled SMD values were not substantially modified when individual studies were sequentially removed (effect size range between 0.52 and 0.91, Figure 3).

#### 3.2.4. Publication Bias

There was no publication bias according to the Begg’s test (*p* = 0.66), the Egger’s test (*p* = 0.24), or the “trim-and-fill” method.

#### 3.2.5. Sub-Group Analysis

Circulating GPx concentrations were significantly higher after statin treatment in the studies assessing whole blood/erythrocytes (SMD = 1.24, 95% CI 0.22 to 2.26, *p* = 0.017; I^2^ = 97.5%, *p* < 0.001, Figure 4A), but not in those assessing serum/plasma (SMD = 0.20, 95% CI −0.34 to 0.74, *p* = 0.463; I^2^ = 80.0%, *p* < 0.001). Specifically, GPx concentrations post-treatment were significantly higher in the studies assessing erythrocytes (SMD = 1.24, 95% CI 0.22 to 2.26, *p* = 0.021; I^2^ = 98.1%, *p* < 0.001, Figure 4B), but not in those assessing whole blood (SMD = 0.24, 95% CI −0.11 to 0.59, *p* = 0.174; I^2^ = 97.5%, *p* < 0.001). In studies assessing erythrocytes, the SMD with individual statins (fluvastatin, simvastatin, atorvastatin) was similar (Figure 5).

#### 3.2.6. Certainty of Evidence

The initial level of certainty for GPx SMD values was moderate as the studies were interventional (rating 3, ⊕⊕⊕⊝). As 11 out of 13 studies had a low risk of bias (no rating change required), there was extreme and unexplained heterogeneity (serious limitation, downgrade one level), there was a lack of indirectness (no rating change required), the imprecision was low (narrow confidence intervals without threshold crossing, upgrade one level), the effect size was large (SMD = 0.80, upgrade one level), and there was no publication bias (no rating change required), the overall level of certainty was considered high (rating 4, ⊕⊕⊕⊕).

### 3.3. Superoxide Dismutase

#### 3.3.1. Study Characteristics

A total of 8 studies, reporting 10 treatment arms in 542 patients (mean age 53 years, 52% males), presented data on SOD [27,49,50,51,53,54,56,59]. Four studies (five arms) assessed erythrocytes [27,50,51,54], two serum [49,56], and one whole blood [53] and plasma [59], respectively. The statin used was atorvastatin in four studies [49,50,51,59], simvastatin in two [51,56], fluvastatin in two [27,54], and rosuvastatin in one [53]. The treatment duration ranged between 4 and 24 weeks (Table 1).

#### 3.3.2. Risk of Bias

The risk of bias was low in all studies [27,49,50,51,53,54,56,59] (Table 2).

#### 3.3.3. Results of Individual Studies and Syntheses

The forest plot of the circulating SOD concentrations before and after statin treatment is shown in Figure 6. In all treatment arms, SOD concentrations were higher after statin treatment (mean difference range 0.27 to 6.95), with significant differences reported in seven arms [50,51,54,56,59]. In view of the extreme heterogeneity observed (I^2^ = 96.9%, *p* < 0.001), random-effects models were used. Pooled results showed that the circulating SOD concentrations were significantly higher after statin treatment (SMD = 1.54, 95% CI 0.71 to 2.36, *p* < 0.001). In the sensitivity analysis, the pooled SMD values were not modified when individual studies were omitted (effect size range between 0.98 and 1.69, Figure 7).

#### 3.3.4. Publication Bias

No publication bias was observed with either the Begg’s test (*p* = 0.72), the Egger’s test (*p* = 0.61), or the “trim-and-fill” method.

#### 3.3.5. Sub-Group Analysis

Post-treatment SOD concentrations were significantly higher both in studies assessing whole blood/erythrocytes (SMD = 1.97, 95% CI 0.54 to 3.40, *p* < 0.001; I^2^ = 98.1%, *p* < 0.001, Figure 8A) and in those assessing serum/plasma (SMD = 0.95, 95% CI 0.47 to 1.44, *p* < 0.001; I^2^ = 75.0%, *p* < 0.001). Non-significant differences in SMD (t = −3.33, *p* = 0.08) were observed between studies measuring SOD in plasma (SMD = 1.32, 95% CI 1.08 to 1.56, *p* < 0.001, Figure 8B) and those assessing serum (SMD = 0.47, 95% CI 0.02 to 0.91, *p* = 0.038). In both cases, however, no heterogeneity was observed (I^2^ = 0.0%). The SMD with individual statins (fluvastatin, simvastatin, atorvastatin) was similar (Figure 9).

#### 3.3.6. Certainty of Evidence

The initial certainty for the SOD SMD values was moderate (interventional studies; rating 3, ⊕⊕⊕⊝). The final level of certainty was high (rating 4, ⊕⊕⊕⊕) due to the low risk of bias in all studies (no rating change), the extreme and unexplained heterogeneity (downgrade one level), the lack of indirectness (no rating change), the low imprecision (upgrade one level), the large effect size (SMD = 1.54, upgrade one level), and the absence of publication bias (no rating change).

### 3.4. Catalase

#### 3.4.1. Study Characteristics

A total of 3 studies, reporting 4 treatment arms in 61 patients (mean age 51 years, 63% males), presented data on serum catalase [49,56,58]. The statin used was atorvastatin in two studies [49,58], and simvastatin in the remaining one [56]. The treatment duration ranged between 4 and 24 weeks (Table 1).

#### 3.4.2. Risk of Bias

The risk of bias was considered low in all studies [49,56,58] (Table 2).

#### 3.4.3. Results of Individual Studies and Syntheses

The forest plot of the circulating catalase concentrations before and after statin treatment is shown in Figure 10. The catalase concentrations increased in one study [49], and decreased in the other two [56,58]. However, in no study was a significant difference reported. Accordingly, the pooled results showed that the circulating catalase concentrations did not significantly change after statin treatment (SMD = −0.16, 95% CI −0.51 to 0.20, *p* = 0.391). There was a low between-study heterogeneity (I^2^ = 0.00%, *p* = 0.391). In the sensitivity analysis, the corresponding pooled SMD values were not substantially modified when individual studies were sequentially omitted (effect size range between −0.29 and 0.00, Figure 11).

#### 3.4.4. Publication Bias

An assessment of publication bias was not possible due to the limited number of studies.

#### 3.4.5. Sub-Group Analysis

A sub-group analysis was not possible due to the limited number of studies.

#### 3.4.6. Certainty of Evidence

The initial level of certainty was moderate as the studies were interventional (rating 3, ⊕⊕⊕⊝). This was downgraded to very low (rating 0, ⊝⊝⊝⊝) after considering the low risk of bias in all studies (no change), the low heterogeneity (no change), the lack of indirectness (no change), the high imprecision (downgrade one level), the small effect size (SMD = −0.16, downgrade one level), and the lack of assessment of publication bias (downgrade one level).

## 4. Discussion

Statins significantly increased the circulating concentrations of the antioxidant enzymes GPx and SOD, but not catalase, in patients with various cardiovascular risk burdens. The observed SMD values for GPx (0.80) and SOD (1.54) suggest a large effect size, and therefore the presence of tangible antioxidant effects [33]. Furthermore, in sensitivity analysis, the corresponding pooled SMDs were not substantially modified when individual studies were sequentially removed. Importantly, the certainty of evidence was considered high for both GPx and SOD.

The presence of hypercholesterolemia, singly or in combination with other traditional cardiovascular risk factors, favours the production of ROS by NADPH oxidase, xanthine oxidase, the mitochondrial electron-transport chain, and uncoupled nitric oxide synthase [60,61]. This, in combination with an impaired function of key antioxidant systems that include GPx, SOD, and catalase, promotes oxidative stress and, consequently, the development of endothelial dysfunction, vascular damage, and atherosclerosis [30,62]. The main biological effects of GPx, SOD, and catalase are well established. GPx catalyses the reduction of free H_2_O_2_, a precursor of the highly reactive radical OH^•^, to H_2_O and their corresponding alcohols. While eight isoforms of GPx have been reported in humans (GPx1-8), GPx1 is the most abundant and commonly measured isoform [63]. The three isoforms of SOD (SOD1-3) catalyse the dismutation of the superoxide anion, O_2_^−•^, into O_2_ and H_2_O_2_ [64]. By contrast, catalase, a tetramer of four polypeptides, promotes the transformation of H_2_O_2_ into O_2_ and H_2_O [65]. The key pathophysiological role of GPx, SOD, and catalase in human atherosclerosis was highlighted in a systematic review and meta-analysis of 3 cohort and 41 case–control studies. The pooled odds ratio for coronary heart disease was significantly and inversely associated with a 1-standard deviation increase in GPx (0.51, 95% CI 0.35 to 0.75), SOD (0.48, 95% CI 0.32 to 0.72), and catalase (0.32, 95% CI 0.16 to 0.61) [31]. The associations with GPx and SOD were similar in patients with acute and chronic coronary heart disease. By contrast, the associations with catalase were stronger in patients with acute coronary heart disease [31].

Our meta-analysis supports a significant antioxidant effect of statins through the upregulation of GPx and SOD. The absence of tangible effects of statin treatment on the concentrations of catalase needs to be interpreted with caution due to the small number of eligible studies identified (n = 3). While these data are encouraging in terms of atheroprotection, the exact mechanisms involved in the statin-mediated upregulation of antioxidant enzymes require additional in vitro and in vivo studies. Furthermore, appropriately designed interventional studies are warranted to determine whether the beneficial effects of this class of drugs in terms of primary and secondary cardiovascular prevention are, at least partly, mediated by specific antioxidant effects that are independent of cholesterol lowering.

The relatively small number of studies identified for analysis and the extreme between-study heterogeneity when reporting GPx and SOD concentrations represent the significant limitations of our study. However, virtually no heterogeneity was observed in a subgroup of studies investigating SOD concentrations specifically in serum of plasma. Additional limitations include the different biological matrices used for the assessment of GPx and SOD, and the lack of serial assessment of these enzymes throughout the treatment period. Significant strengths include the lack of publication bias and the high certainty of evidence with GPx and SOD, suggesting that the effect of statins on these enzymes is both genuine and biologically plausible.

## 5. Conclusions

Statin treatment significantly increases the circulating concentrations of the antioxidant enzymes GPx and SOD using a range of biological matrices, suggesting the protective effects of these agents against oxidative stress. Intervention studies are warranted to investigate the antioxidant effects of statins as important mediators of their beneficial effects for primary and secondary cardiovascular prevention, to determine the most suitable biological matrix for GPx and SOD assessment, and to identify specific patient groups that are more likely to benefit from a combined antioxidant and the lipid-lowering effect of this class of drugs.

## Figures and Tables

**Figure 1 antioxidants-10-01841-f001:**
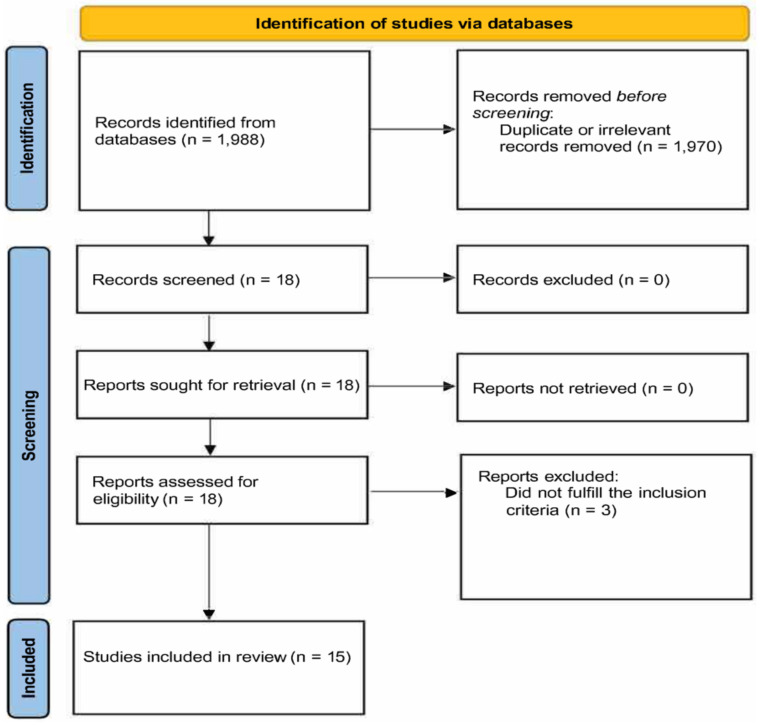
PRISMA 2020 flow diagram.

**Figure 2 antioxidants-10-01841-f002:**
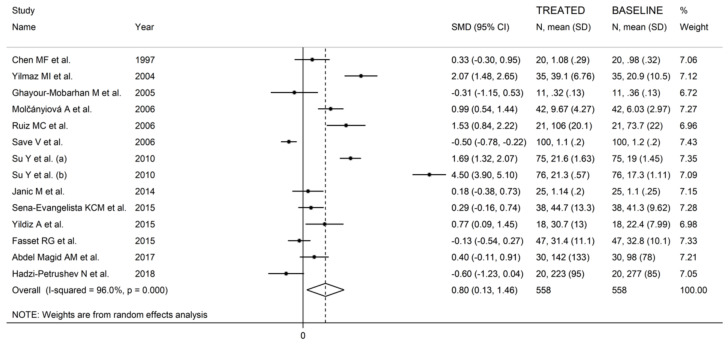
Forest plot of studies reporting GPx concentrations before and after statin treatment.

**Figure 3 antioxidants-10-01841-f003:**
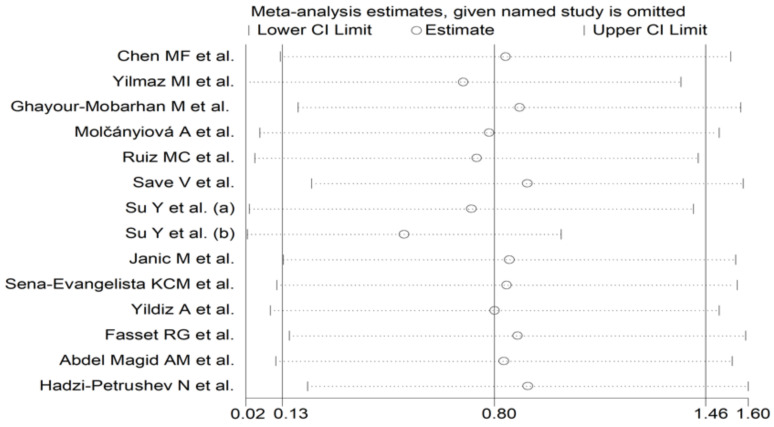
Influence of individual studies on the standardized mean difference (SMD). The hollow circles represent the SMD when the remaining study is omitted.

**Figure 4 antioxidants-10-01841-f004:**
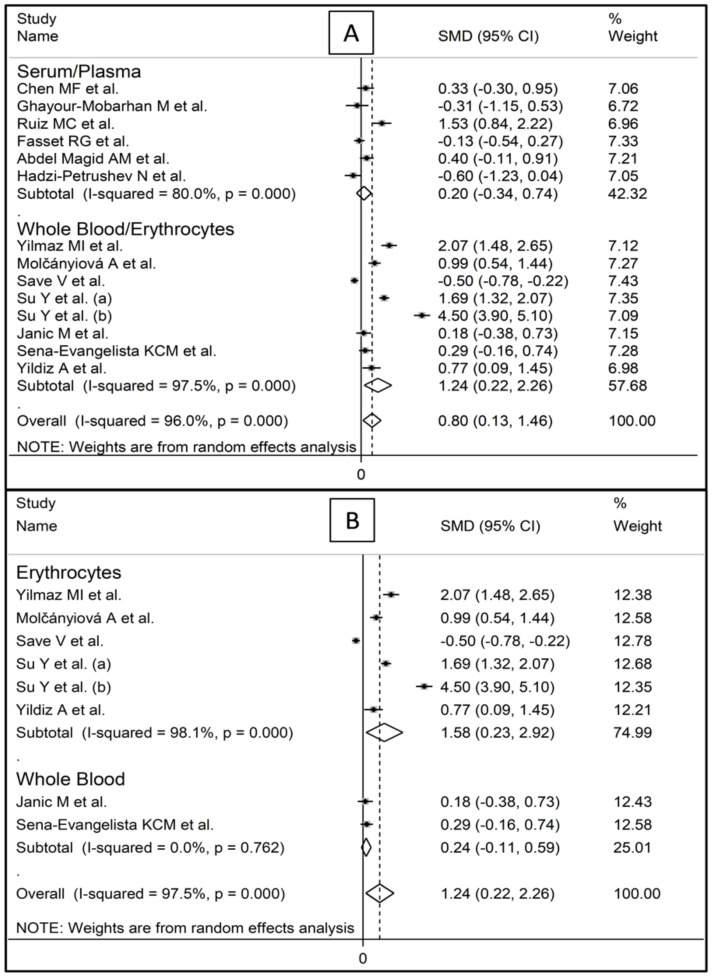
Forest plot of studies investigating GPx concentrations according to biological matrix: (**A**) whole blood/erythrocytes vs. plasma/serum; (**B**) whole blood vs. erythrocytes.

**Figure 5 antioxidants-10-01841-f005:**
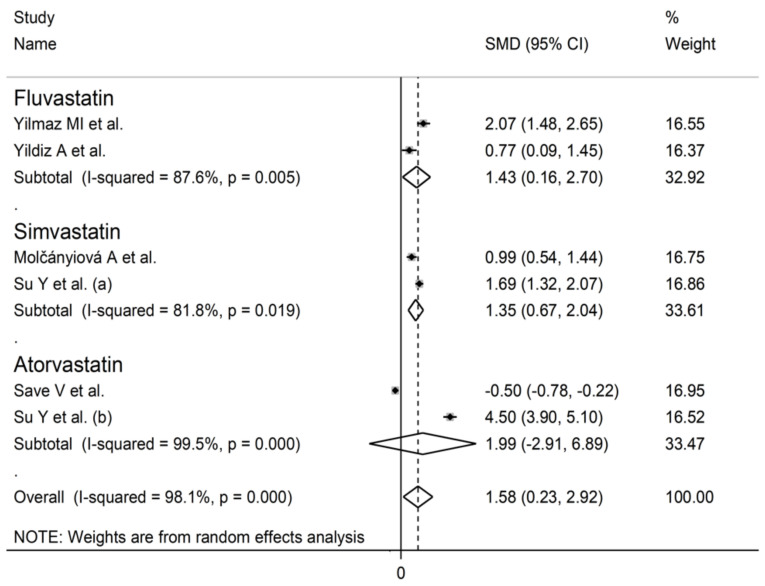
Forest plot of studies of individual statins on GPx concentrations.

**Figure 6 antioxidants-10-01841-f006:**
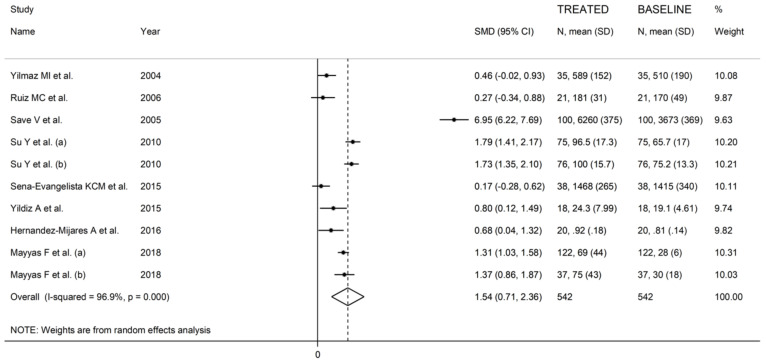
Forest plot of SOD concentrations before and after statin treatment.

**Figure 7 antioxidants-10-01841-f007:**
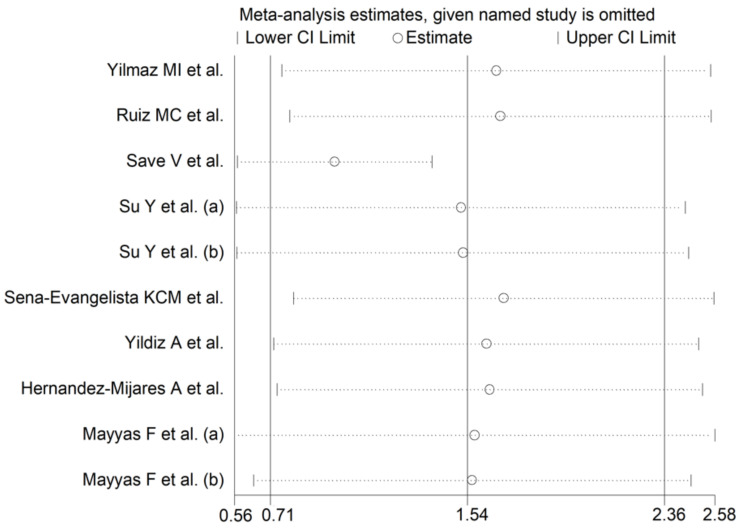
Sensitivity analysis describing the impact of individual studies on SOD on the standardized mean difference.

**Figure 8 antioxidants-10-01841-f008:**
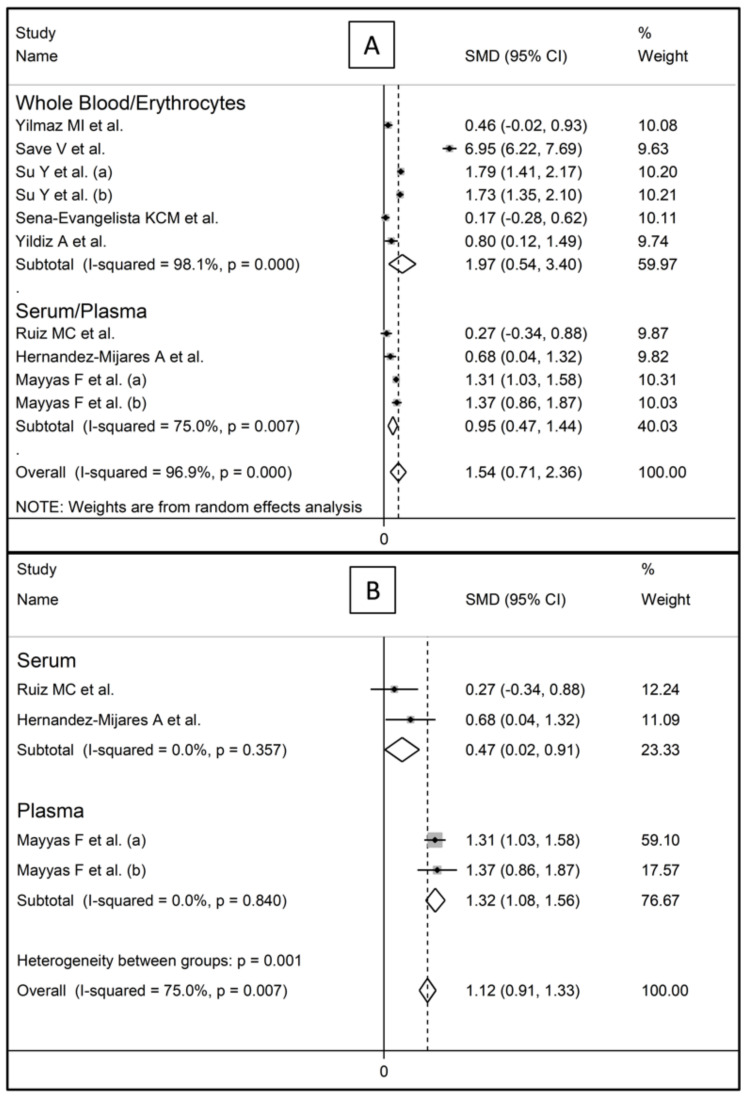
Forest plot of studies investigating SOD concentrations according to the biological matrix: (**A**) whole blood/erythrocytes vs. plasma/serum; (**B**) plasma vs. serum.

**Figure 9 antioxidants-10-01841-f009:**
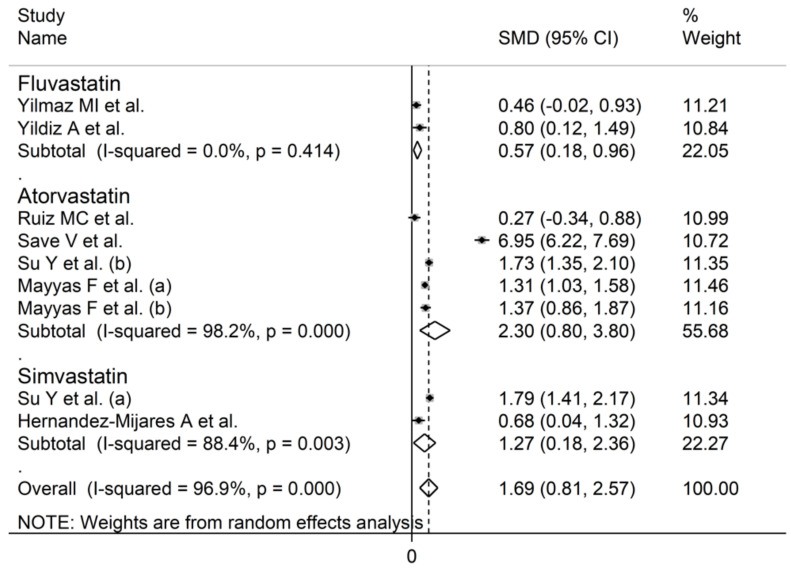
Forest plot of studies of individual statins on SOD concentrations.

**Figure 10 antioxidants-10-01841-f010:**
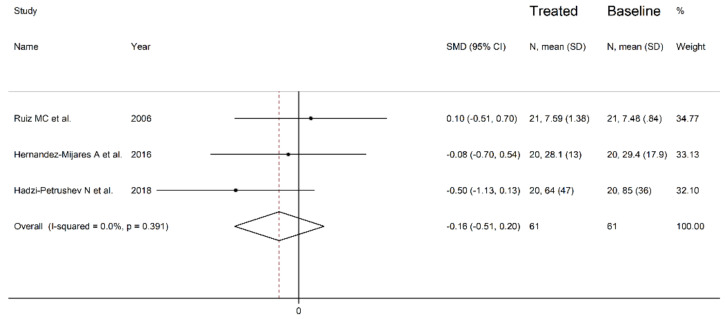
Forest plot of catalase concentrations before and after statin treatment.

**Figure 11 antioxidants-10-01841-f011:**
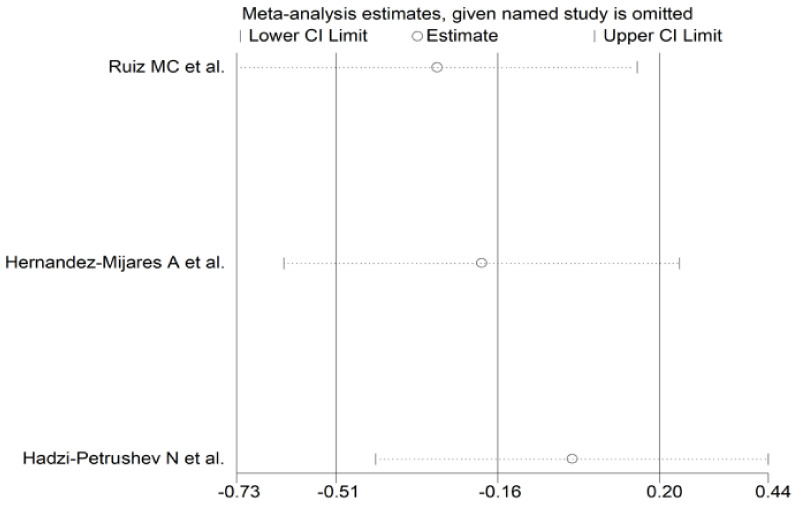
Sensitivity analysis of the influence of each study on the overall standardized mean difference.

**Table 1 antioxidants-10-01841-t001:** Study characteristics.

First Author, Year, Country [Ref]	Matrix	n	Age (yrs)	M/F	GPx Bas Mean ± SD	GPx Post Mean ± SD	SOD Bas Mean ± SD	SOD Post Mean ± SD	Cat Bas Mean ± SD	Cat Post Mean ± SD	Condition	Statin and Daily Dose	Treatment (weeks)
Chen MF, 1997, Taiwan [46]	P	20	47	11/9	0.98 ± 0.32 U/mL	1.08 ± 0.29 U/mL	NR	NR	NR	NR	HCL	Pravastatin 5 mg	12
Yilmaz MI, 2004, Turkey [27]	E	35	48	18/17	20.93 ± 10.46 U/mL	39.13 ± 6.76 U/mL	510 ± 190 U/mL	589 ± 182 U/mL	NR	NR	HCL	Fluvastatin 40 mg	12
Ghayour-Mobarhan M, 1997, UK [47]	S	11	52	7/4	0.36 ± 0.13 U/mL	0.32 ± 0.13 U/mL	NR	NR	NR	NR	HCL	Simvastatin 10 mg	16
Molčányiová A, 2006, Slovakia [48]	E	42	60	12/30	6.03 ± 2.97 U/mL	9.67 ± 4.27 U/mL	NR	NR	NR	NR	HCL	Simvastatin 20 mg	8
Ruiz MC, 2006, Spain [49]	S	21	NR	NR	74 ± 22 nmol/mg	106 ± 22 nmol/mg	170 ± 49 U/mg	181 ± 31 U/mg	7.48 ± 0.84 KU × 10^−5^/mg	7.59 ± 1.38 KU × 10^−5^/mg	Kidney Tx	Atorvastatin 10–40 mg	24
Save V, 2006, India [50]	E	100	51	29/71	1.2 ± 0.2 U/mL	1.1 ± 0.2 U/mL	3673 ± 369 U/gHb	6260 ± 375 U/gHb	NR	NR	T2D	Atorvastatin 10 mg	24
Su Y, 2010 (a), China [51]	E	75	55	39/36	18.96 ± 1.45 µmol/L	21.57 ± 1.63 µmol/L	65.73 ± 17.02 mmol/L	96.54 ± 17.34 mmol/L	NR	NR	T2D	Simvastatin 40 mg	12
Su Y, 2010 (b), China [51]	E	76	56	43/33	17.31 ± 1.11 µmol/L	21.28 ± 0.57 µmol/L	75.15 ± 13.31 mmol/L	100.23 ± 15.67 mmol/L	NR	NR	T2D	Atorvastatin 10 mg	12
Janic M, 2014, Slovenia [52]	WB	25	44	25/0	1.10 ± 0.25 U/gHb	1.14 ± 0.20 U/gHb	NR	NR	NR	NR	Healthy	Fluvastatin 10 mg	4.5
Sena-Evangelista KCM, 2015, Brazil [53]	WB	38	63	23/15	41.33 ± 9.62 U/gHb	44.67 ± 13.33 U/gHb	1415 ± 340 U/gHg	1468 ± 265 U/gHg	NR	NR	CAD	Rosuvastatin 10 mg	16
Yildiz A, 2015, Turkey [54]	E	18	38	9/9	22.37 ± 7.99 U/gHb	30.7 ± 13.4 U/gHb	19.09 ± 4.61 U/gHg	24.34 ± 7.99 U/gHg	NR	NR	Kidney Tx	Fluvastatin 80 mg	4
Fassett RG, 2015, Australia [55]	P	47	65	28/19	32.8 ± 10.1 U/L	31.4 ± 11.1 U/L	NR	NR	NR	NR	CKD	Atorvastatin 10 mg	3 years
Hernandez-Mijares A, 2016, Spain [56]	S	20	58	5/15	NR	NR	0.81 ± 0.14 U/mL	0.92 ± 0.18 U/mL	29.4 ± 17.9 U/mL	28.1 ± 13.0 U/mL	HCL	Simvastatin 40 mg	4
Abdel Magid AM, 2017, Egypt [57]	S	30	51	15/15	98 ± 78 U/L	142 ± 133 U/L	NR	NR	NR	NR	HD	Simvastatin 60 mg *	16
Hadzi-Petrushev N, 2018, Macedonia [58]	S	20	43	20/0	277 ± 85 U/mL	223 ± 95 U/mL	NR	NR	85 ± 36 U/mL	64 ± 47 U/mL	NAFLD	Atorvastatin 20 mg	12
Mayyas F, 2018 (a), Jordan [59]	P	122	51	81/41	NR	NR	28 ± 6 U/mL	69 ± 44 U/mL	NR	NR	ASCVD	Atorvastatin 20 mg	12
Mayyas F, 2018 (b), Jordan [59]	P	37	51	24/13	NR	NR	30 ± 18 U/mL	75 ± 43 U/mL	NR	NR	ASCVD	Atorvastatin 40 mg	12

Legend: P, plasma; S, serum; E, erythrocytes; WB, whole blood; GPx, glutathione peroxidase; SOD, superoxide dismutase; Cat, catalase; HCL, hypercholesterolemia; Tx, transplant; T2D, type 2 diabetes; CAD, coronary artery disease; CKD, chronic kidney disease; HD, haemodialysis; NAFLD, non-alcoholic fatty liver disease; ASCVD, atherosclerotic cardiovascular disease; NR, not reported; *, weekly.

**Table 2 antioxidants-10-01841-t002:** The Joanna Briggs Institute Critical Appraisal Checklist.

Study	Were the Criteria for Inclusion in the Sample Clearly Defined?	Were the Study Subjects and the Setting Described in Detail?	Was the Exposure Measured in a Valid and Reliable Way?	Were Objective, Standard Criteria Used for Measurement of the Condition?	Were Confounding Factors Identified?	Were Strategies to Deal with Confounding Factors Stated?	Were the Outcomes Measured in a Valid and Reliable Way?	Was Appropriate Statistical Analysis Used?	Risk of Bias
Chen MF [46]	Yes	Yes	Yes	Yes	No	No	Yes	No	Low
Yilmaz MI [27]	Yes	Yes	Yes	Yes	No	No	Yes	No	Low
Ghayour-Mobarhan M [47]	No	No	Yes	Yes	No	No	Yes	No	High
Molčányiová A [48]	Yes	Yes	Yes	Yes	Yes	Yes	Yes	Yes	Low
Ruiz MC [49]	Yes	Yes	Yes	Yes	No	No	Yes	No	Low
Save V [50]	Yes	Yes	Yes	Yes	No	No	Yes	No	Low
Su Y [51]	Yes	Yes	Yes	Yes	No	No	Yes	No	Low
Janic M [52]	No	No	Yes	Yes	No	No	Yes	No	High
Sena-Evangelista KCM [53]	Yes	Yes	Yes	Yes	No	No	Yes	No	Low
Yildiz A [54]	Yes	Yes	Yes	Yes	No	No	Yes	No	Low
Fassett RG [55]	Yes	Yes	Yes	Yes	No	No	Yes	No	Low
Hernandez-Mijares A [56]	Yes	Yes	Yes	Yes	No	No	Yes	No	Low
Abdel Magid AM [57]	Yes	Yes	Yes	Yes	No	No	Yes	No	Low
Hadzi-Petrushev N [58]	Yes	Yes	Yes	Yes	No	No	Yes	No	Low
Mayyas F [59]	Yes	Yes	Yes	Yes	No	No	Yes	No	Low

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
