# Peer review of "A Systematic Review and Meta-Analysis of the Effect of Statins on Glutathione Peroxidase, Superoxide Dismutase, and Catalase"

_antioxidants, 2021, doi:10.3390/antiox10111841_

Round 1

Reviewer 1 Report

The antioxidative effect of statin has been extensively studied in atherosclerosis. The authors performed meta-analyses of studies reporting the effects of statin treatment on the circulating concentrations of GPx, SOD, and catalase in the patients with different cardiovascular risk profiles. Their meta-analyses support the hypothesis that statin would increase these enzyme concentrations in blood of patients receiving statin treatment.

Although better understanding of the mechanism of statin action will impact of the clinical strategy, the contribution of this study may be limited and slightly out of the scope of this journal.

Major Concerns:

Antioxidant effects of statins has been suggested and studied. Although this manuscript supports the concept in an indirect way, not much insight would be added about the mechanisms of statin action.

In general, the level of enzyme markers in blood can be increased in many ways (de novo synthesis, extension of stay, cell damage-based release…) and due to the various sources, such as vascular endothelium and erythrocytes. Since the basis for relying on underlying molecular mechanism is ambiguous, the objective of this research is unclear. Indeed, this study ignores to discuss the other anti-oxidative pathways have been suggested as of statin’s pleiotropic effects, such as eNOS and NADPH oxidase.

Author Response

Thank you for your feedback. Our responses to the points raised are as follows:

Antioxidant effects of statins has been suggested and studied. Although this manuscript supports the concept in an indirect way, not much insight would be added about the mechanisms of statin action. In general, the level of enzyme markers in blood can be increased in many ways (de novo synthesis, extension of stay, cell damage-based release…) and due to the various sources, such as vascular endothelium and erythrocytes. Since the basis for relying on underlying molecular mechanism is ambiguous, the objective of this research is unclear. Indeed, this study ignores to discuss the other anti-oxidative pathways have been suggested as of statin’s pleiotropic effects, such as eNOS and NADPH oxidase.

We agree about the complexity of the proposed pleiotropic effects of statins. However, we also wish to highlight that 1) the objective of our systematic review and meta-analysis was not to identify or confirm the molecular mechanisms involved, rather to critically appraise the available evidence on the effect of statin treatment on the selected antioxidant enzymes, 2) conducting additional analyses on other pathways, e.g. eNOS and NADPH oxidase, would require a separate manuscript, and 3) the additional anti-oxidative effects of statins are discussed in the original introduction:

“In particular, statins have been shown to inhibit key pro-oxidant enzymes such as nicotinamide adenine dinucleotide phosphate (NADPH) oxidase [18,19], reduce the synthesis of the highly reactive compound malondialdehyde from lipid peroxidation of polyunsaturated fatty acids [20], as well as upregulate antioxidant enzymes such as catalase [21], glutathione peroxidase (GPx) [22], and superoxide dismutase (SOD) [23-25].”

This sentence has been amended to include the effects on eNOS, as follows:

“In particular, statins have been shown to inhibit key pro-oxidant enzymes such as nicotinamide adenine dinucleotide phosphate (NADPH) oxidase [18,19], reduce the synthesis of the highly reactive compound malondialdehyde from lipid peroxidation of polyunsaturated fatty acids [20], as well as increase the expression, activity and coupling of endothelial nitric oxide synthase [21], and upregulate antioxidant enzymes such as catalase [22], glutathione peroxidase (GPx) [23], and superoxide dismutase (SOD) [24-26].”

Reviewer 2 Report

This systematic mini-review provides readers a brief overview of the effect of statins associated with anti-oxidant activity. Authors pointed out that antioxidant enzymes such as glutathione peroxidase, superoxide dismutase, and catalase were increased after statin supplementation. Introduction, methods, and discussion are written very clearly. Results are presented in illustrative tables and figures.
The limitation of this review is the relatively small number of studies and the study heterogeneity, which also authors themselves emphasize in the discussion part.

Author Response

This systematic mini-review provides readers a brief overview of the effect of statins associated with anti-oxidant activity. Authors pointed out that antioxidant enzymes such as glutathione peroxidase, superoxide dismutase, and catalase were increased after statin supplementation. Introduction, methods, and discussion are written very clearly. Results are presented in illustrative tables and figures. The limitation of this review is the relatively small number of studies and the study heterogeneity, which also authors themselves emphasize in the discussion part.

Thank you for the positive feedback.

Reviewer 3 Report

In the present systemic review and meta-analysis, the authors mainly have reviewed to explore the protective effects of statins in arresting reactive oxygen-sensing mechanisms by the upregulation of the circulating level of antioxidant enzymes. According to their review, administration of statins among the patients resulted in increased expression for glutathione peroxidase and superoxide dismutase, but they didn't find any change with the catalase enzyme. However, the manuscript has been written well with an adequate write-up in the methodology, result, and discussion section as well. Statistical analysis is also adequate. This manuscript can be accepted only after a few minor corrections. I would like to suggest the authors to add a few more relevant references in the second paragraph of the introduction section where they stated the role of oxidative stress, prooxidants, and LDL in the development of atherosclerosis and proinflammatory responses. Further, the authors need to add a few more relevant references in the 2nd paragraph of the discussion section to discuss on the interplay between hypercholesterolemia, reactive oxygen species, and antioxidant enzymes in the context of atherosclerosis. 

Author Response

Many thanks for your feedback. Our responses to the points raised are as follows:

I would like to suggest the authors to add a few more relevant references in the second paragraph of the introduction section where they stated the role of oxidative stress, prooxidants, and LDL in the development of atherosclerosis and proinflammatory responses. Further, the authors need to add a few more relevant references in the 2nd paragraph of the discussion section to discuss on the interplay between hypercholesterolemia, reactive oxygen species, and antioxidant enzymes in the context of atherosclerosis. 

As requested, the following additional relevant references have been added in both paragraphs:

[12]     Marchio P, Guerra-Ojeda S, Vila JM, Aldasoro M, Victor VM, Mauricio MD. Targeting Early Atherosclerosis: A Focus on Oxidative Stress and Inflammation. Oxid Med Cell Longev 2019; 2019: 8563845. doi: 10.1155/2019/8563845

[13]     Ahotupa M. Oxidized lipoprotein lipids and atherosclerosis. Free Radic Res 2017; 51: 439-447. doi: 10.1080/10715762.2017.1319944

[14]     Khatana C, Saini NK, Chakrabarti S, et al. Mechanistic Insights into the Oxidized Low-Density Lipoprotein-Induced Atherosclerosis. Oxid Med Cell Longev 2020; 2020: 5245308. doi: 10.1155/2020/5245308

[17]     Yang X, Li Y, Li Y, et al. Oxidative Stress-Mediated Atherosclerosis: Mechanisms and Therapies. Front Physiol 2017; 8: 600. doi: 10.3389/fphys.2017.00600

[60]     Oliveira HCF, Vercesi AE. Mitochondrial bioenergetics and redox dysfunctions in hypercholesterolemia and atherosclerosis. Mol Aspects Med 2020; 71: 100840. doi: 10.1016/j.mam.2019.100840

[61]     Sozen E, Ozer NK. Impact of high cholesterol and endoplasmic reticulum stress on metabolic diseases: An updated mini-review. Redox Biol 2017; 12: 456-461. doi: 10.1016/j.redox.2017.02.025

[62]     Thomas SR, Witting PK, Drummond GR. Redox control of endothelial function and dysfunction: molecular mechanisms and therapeutic opportunities. Antioxid Redox Signal 2008; 10: 1713-1765. doi: 10.1089/ars.2008.2027

Reviewer 4 Report

In this systematic review and meta-analysis of the fifteen selected clinical studies, the author concisely and comprehensively reported the effects of statin treatment on antioxidant enzymes(glutathione peroxidase ;GPx, superoxide dismutase ;SOD, and catalase) in patients with various clinical back ground (hypercholestemia, type 2 diabetes, NAFLD and atherosclerotic caridiovascular diseases).

In 15 studies(17 treatment arms in 773 patients), statins significantly increased both GPx and SOD but not catalase.  But in the study of catalase, only three studies, reporting four treatment arms in 61 patients were analyzed and effect size was small and the level of certainty was very low.

  The study design was reasonable.

The methods of statistical analyses were appropriate.

1) Risk of bias was assessed with Joanna Briggs Institute Critical Appraisal Checklist for analytical studies.

2) Certainty of evidence was assessed using GRADE.

3) Between study heterogeneity was assessed with the Q-static and Is static.

4) For investigation of the influence of each study on the overall risk estimate, sensitivity analysis was performed by sequentially removing individual studies.

5) Publication bias was assessed using Begg’s adjusted rank correlation test, The Egger’s regression asymmetry test and Tweedie “ trim- and – fill” procedure.

In results, it was important finding that the different biomaterial matrices (red blood cells, white bood cells, palsma and serum) affected the resuts of post-treatment GPx changes.

Although the relatively small number of studies were analyzed, it was also interesting that subgroup analysis of individual statins in GPx and SOD, the SMD was similar. As various statins (atorvastain, simvastatin, fluvastatin, rosuvastatin and pravastatin) have different lipid lowering effects and chemical properties, so the putative “class” effects of statins to promote antioxidant enzymes should be explored in future well designed studies.

This article is appropriate for publication.

Author Response

Although the relatively small number of studies were analyzed, it was also interesting that subgroup analysis of individual statins in GPx and SOD, the SMD was similar. As various statins (atorvastain, simvastatin, fluvastatin, rosuvastatin and pravastatin) have different lipid lowering effects and chemical properties, so the putative “class” effects of statins to promote antioxidant enzymes should be explored in future well designed studies. This article is appropriate for publication.

Thank you for the positive feedback.

Round 2

Reviewer 1 Report

The concerns have been addressed in the revised manuscript.